# Nomenclature Survey of the Genus *Amaranthus* (Amaranthaceae): 12 Questions about *Amaranthus caudatus*

**DOI:** 10.3390/plants12071566

**Published:** 2023-04-05

**Authors:** Duilio Iamonico

**Affiliations:** Department of Environmental Biology, University of Rome Sapienza, Piazzale Aldo Moro 5, 00185 Rome, Italy; duilio.iamonico@uniroma1.it

**Keywords:** *Amaranthus baileyanus* nom. nov., *Amaranthus mantegazzianus*, synonymy, typification

## Abstract

Nomenclatural and taxonomic issues concerning *Amaranthus caudatus* and the related taxa are presented. Types are designated for names *A. caudatus* var. *albiflorus* (neotype at RO), *A. caudatus* var. *atropurpureus* (neotype at GH), *A. caudatus* var. *gibbosus* (neotype at RO), *A. dussi* (neotype at NAP), and *A. edulis* (lectotype at LP). Holotypes are indicated for the names *A. caudatus* var. *pseudopaniculatus* f. *oblongipetalus* (EA), *A. caudatus* var. *pseudopaniculatus* f. *pseudopaniculatus* (EA), *A. caudatus* subsp. *saueri* (PR), and *Amaranthus edulis* var. *spadiceus* (CORD). The names *A. caudatus* var. *albiflorus*, *A. caudatus* var. *atropurpureus*, *A. caudatus* subsp. *saueri*, *A. dussi*, and *Amaranthus edulis* var. *spadiceus* are considered as hererotypic synonyms of *A. caudatus*. On the basis of morphological, cytological, and molecular data, the taxa caudatus, mantegazzianus, and gibbosus are here proposed to be treated as different species. A new name—*Amaranthus baileyanus*—is proposed for *A. caudatus* var. gibbosus because of a previous and validly published *Amaranthus gibbosus*.

## 1. Introduction

*Amaranthus* L. (Amaranthaceae Juss.) is a genus comprising 70–75 species, of which approximately half are native to the Americas [1,2]. Several American species are used as ornamentals, food, and medicines, and some of them are able to escape from cultivation, mainly impacting agricultural systems economically with reductions in productivity and crop quality [1,2,3,4].

*Amaranthus* is a critical genus from a taxonomical point of view because of its high phenotypic variability, which led to nomenclatural disorders and misapplication of names [1,2,5,6]. No comprehensive molecular study has been published at present yet, and, on the basis of the more recent classification [5], three subgenera were recognized: subgenus *Acnida* (L.) Aellen ex K.R. Robertson with three sections, subgenus *Albersia* (Kunth) Gren. & Godr. with four sections, and subgenus *Amaranthus*, with three sections and two subsections. Note, however, that the most recent molecular investigation [7] showed that the classification proposed by Mosyakin and Robertson [5] cannot be retained at the current state of knowledge.

As part of the ongoing study on the nomenclature of all of the published *Amaranthus* names, I here present the twelfth contribution; the previous papers were on the Linnean names [8,9], the names linked to the Italian flora [10], *Amaranthus gracilis* Desf. and related names [11], Moquin-Tandon’s names [12], names linked to the Australian flora [13], Willdenow’s names [14], *Amaranthus polygonoides* L. *s.l.* [15], Roxburgh’s names [16], *A. commutatus* A.Kern [17], and members of the subgen. Acnida (L.) Aellen ex K.R.Robertson sensu Mosyakin and Robertson [18].

## 2. Material and Methods

This work is based on field surveys, analysis of relevant literature (protologues are included), and checking/examination of specimens preserved in the following herbaria: BH, BM, BR, CFL, CORD, EA, FI, GH, HAL, HOH, K, LINN, LP, MEL, M, MO, NAP, NY, P, PH, RO, SI, UCBD, and US (acronyms according to Thiers [19]).

The articles cited throughout the text are referred to the *Shenzhen Code* (hereafter reported as “ICN” [20]).

## 3. Results and Discussion

### 3.1. Nomenclatural Notes

#### 3.1.1. *Amaranthus caudatus* subsp. *saueri*

*Amaranthus caudatus* subsp. *saueri* was described by Jehlik in 1990 [21] to distinguish forms characterized in having rose- or white-coloured seeds with an obtuse margin (diagnosis: “Semina rosacea usque fere albicantia, margine obtusa”), whereas the autonymic species was recognized in displaying dark seeds. The holotype is preserved at PR (barcode PR615740). I had the opportunity to examine this specimen (Figure 1) and observed the colour of the seeds, which varies from rose (light brownish in exsiccatum) to white. However, the colour of the seeds cannot be considered at present as a character which allows to distinguish infraspecific ranks and this variation in colour is currently included in the variability of *A. caudatus* as reported by several authors (see, e.g., [1,2,6,22,23]). *A. caudatus* subsp. *saueri* does not deserve to be considered as a separate taxon from *A. caudatus s.lat.* and it is a synonym of *A. caudatus s.s.*, having the pendulous terminal florescence.

#### 3.1.2. *Amaranthus caudatus* var. *alopecurus*

*Amaranthus caudatus* var. *alopecurus* was validly published by Moquin-Tandon as part of his treatment of *Amaranthus* in Candolle’s *Prodromus* [24] (p. 256). The lectotypification of this name was proposed twice by Bajón [6] and Iamonico [12]. Both of the lectotype designations refer to the same specimen deposited at P (barcode P00482809). Isolectotypes were also listed by both of these authors on specimens preserved at GH (barcode GH00037040), HOH (barcode HOH009263), and MO (barcode MO357985). Furthermore, an isolectotype was also reported at BR (barcode BR000832631) and HAL (barcode HAL0110480) by Iamonico [12], or at K (barcode K000223569) by Bajón [6]. Since Bayón’s paper was published before Iamonico’s one (29 December 2015 vs. 7 September 2016), the former designation [6] must be followed according to Art. 9.19 of ICN. Finally, further isolectotypes were found during the present research: they are preserved at K (barcode K000223570) and MEL (barcode MEL2459428). According to the treatment proposed in the present paper, *Amaranthus caudatus* var. *alopecurus* is considered as a synonym of *A. caudatus s.s.*, having the pendulous terminal florescence.

#### 3.1.3. *Amaranthus caudatus* var. *pseudopaniculatus*

Suessenguth [25] (p. 71) proposed to describe the var. *pseudopaniculatus* to distinguish plants of *Amaranthus caudatus* with shortly aristate tepals and highly dense branches of the synflorescence; a collection was also cited (“Tanganyika-Territor., Amani, leg. GREENWAY nr. 993 (Herb. Nairobi)”). A new f. *oblongipetalus* Suesseng. (reported “*oblongopetalus*”, here corrected according to Art. 60.10 of ICN) was also described (according to Art. 26.3 of ICN the f. *caudatus* was automatically established) by a short diagnosis (“Tepala oblonga vel anguste oblonga”) and citing the following collection: “Tanganyika-Territor., Amani 2900 ft., leg. GREENWAY nr. 6155 (Herb. Nairobi)”. Townsend [26] (p. 26) listed a specimen deposited at EA (acronym of the herbarium of the National Museums of Kenya which corresponds to “Herb. Nairobi” as reported by Suessenguth and Merxmüller [25]) as the holotype of the var. *pseudopaniculatus*, and a specimen at K as the isotype of the f. *oblongipetalus* (“Type of var.: Tanzania, Lushoto District, Amani, *Greeway* 993 (EA, holo.!) of forma: Tanzania, Lushoto District, Amani, *Greeway* 6155 (K, iso.!)”). I traced the following three specimens:(1)Greeway’s specimen no. 6154 (herbarium EA; collection number is indicated in the label at the base of the plant). A further label on the bottom-left corner of the sheet reports “EAST AFRICAN AGRICULTURAL RESEARCH STATION HERBARIUM, AMANI | no. *993*”. This specimen refers to var. *pseudopaniculatus s.s.*(2)Greeway’s collection no. 6155 (herbarium EA; collection number is indicated in the label on the centre-left of the sheet). A further label (bottom-left corner of the sheet) reports “EAST AFRICAN AGRICULTURAL RESEARCH STATION HERBARIUM, AMANI | no. *995*”. This specimen refers to var. *pseudopaniculatus* f. *oblongipetalus*.(3)Greeway’s collection no. 6155 (herbarium K, barcode K000195694; no. 6155 is indicated in a label on the bottom-centre of the sheet), titled as “FROM THE HERBARIUM OF THE EAST AFRICAN RESEARCH INSTITUTE, AMANI”. This specimen refers to var. *pseudopaniculatus* f. *oblongipetalus*.

Based on the protologue, Suessenguth [25] clearly indicated for var. *pseudopaniculatus s.s.* and var. *pseudopaniculatus* f. *oblongipetalus* both the number of collections and the herbarium in which they were deposited. I here considered this quotation as an indication of holotypes. Townsend [26] correctly stated that EA no. 6154 is the holotype of the var. *pseudopaniculatus s.s.* (Figure 2), whereas the K specimen is the istoype of var. *pseudopaniculatus* f. *oblongipetalus* because of, as discussed above, the occurrence of the printed label “FROM THE HERBARIUM OF THE EAST AFRICAN RESEARCH INSTITUTE, AMANI”. I traced the holotype of the f. *oblongipetalus*, cited by Suessenguth [25] as the specimen EA no. 6155 (Figure 3).

Townsend [26] synonymized Suessenguth’s variety and form with *Amaranthus cruentus* (sub *A. hybridus* L. subsp. *cruentus* (L.) Thell.). However, the tepals of var. *pseudopaniculatus s.l.* (including f. *oblongipetalus*) are ovato-spathulate and reflexed, whereas in *A. cruentus* tepals are ovato-lanceolate (never spathulate) and always erect (see, e.g., [2]). Therefore, Suessenguth’s variety and form are referable to *A. caudatus s.lat*. According to the treatment proposed in the present paper, types of *A. caudatus* var. *pseudopaniculatus* f. *pseudopaniculatus* and f. *oblongipetalus* are identifiable as *A. mantegazzianus* Passer., having erect terminal florescence.

#### 3.1.4. *Amaranthus caudatus* Varieties Described by Bailey

Bailey [27] (p. 270), in his *The standard cyclopedia of horticulture*, published three varieties under *Amaranthus caudatus*, i.e., var. *albiflorus*, var. *atropurpureus*, and var. *gibbosus*; diagnoses are: “Spikes white or greenish white” (var. *albiflorus*), “Foliage blood-red” (var. *atropurpureus*), and “fls. [flowers] red, clustered in more or less separated fascicles or heads” (var. *gibbosus*). The word “Hort.” (= Hortorum) is reported just after each varietal name and it indicates that the plants were cultivated. Note that Bailey, in his previous (year 1909) *Cyclopedia of American Horticulture* [28] (p. 55), published the name *A. atropurpureus* with the same diagnosis as those given for *A. caudatus* var. *atropurpureus* in *The standard cyclopedia of horticulture* [27] (p. 270). First, Bailey’s *Amaranthus atropurpureus* is illegitimate, being a later homonym of the previous one published by Roxburgh (Art. 53.1 of ICN). Second, Bailey [28] stated for his *A. atropurpureus*: “Problably a form of *A. caudatus*. Peraphs the same as Roxburgh’s *A. atropurpureus* from India”. However, Bailey’s *A. atropurpureus* cannot be the same as Roxburgh’s one, since Bailey [28] clearly indicated that his species (and the previous listed *A. caudatus*) have “*Spikes drooping*”, whereas *A. atropurpureus* Roxb. has erect synflorescence and it is a synonym of *A. tricolor* L. according to Iamonico [16] (pp. 560–561, 563). Anyway, I think that Bailey, in *The standard cyclopedia of horticulture* [27], intended to combine his *A. atropurpureus* at the rank of variety under *A. caudatus*, as supposed by himself in *Cyclopedia of American Horticulture* [28] (p. 55). According to Art. 58.1 of ICN (see Ex. 3), the varietal name is legitimate and to be treated as a replacement name, so typified by the type of *A. atropurpureus* (see Art. 7.4 of ICN); furthermore, the correct citation of the variety is *A. caudatus* var. *atropurpureus* L.H.Bailey (and not *A. caudatus* var. *atropurpureus* “(L.H.Bailey) L.H.Bailey”).

Concerning the original material used by Bailey [27,28] to describe these three varieties, he did not mention any herbarium in which specimens could be deposited. Stafleu and Cowan [29] (p. 94) indicated that the Bailey’s herbarium is preserved at BH, where I found only one sheet of *A. caudatus* (barcode BH275892). This sheet bears the following Bailey’s label (A. M. Stalter, per. comm.): “**GARDEN HERBARIUM** OF CORNELL UNIVERSITY EXPERIMENT STATION [printed]|*Trade Name Amaranthus gibbosus*|*Nich Rochester… July 22 1890* [handwritten] | L. H. BAILEY [printed]”. Two further annotations, directly occurring on the sheet, are “*A. caudatus mna*” (on the left of the sheet, just near the lower leaf), which was probably added by Mabel W. Allen who was here at Cornell in the 1930s or so (A. M. Stalter, per. comm.), and “CYCLOPEDIA OF AMERICAN HORTICULTURE [printed] | *A. paniculatus*” (on the left of the sheet, just above the Bailey’s label), where *A. paniculatus* L. (currently accepted as *A. cruentus* L., see Iamonico [2] (p. 55)) is a note suggesting that the specimen was the base for the description of this species in Bailey’s *Cyclopedia of American Horticulture* [27] (A. M. Stalter, per. comm.). This BH exsiccatum can be identified as *Amaranthus caudatus* (see, e.g., [1,2,6,22,23]) but it cannot be referred to the var. *gibbosus* on the basis of the protologue [28], since it displays continuous synflorescences (not interrupted as indicated in the diagnosis of the var. *gibbosus*). As a consequence, it cannot be considered for the lectotypification purpose of the var. *gibbosus*. No further original material was traced for Bayley taxa and, as a consequence, neotypifications are required under Art. 9.8 of ICN as follows:(1)*Amaranthus atropurpureus*: since the colour of the leaves often change after exsiccation of amaranths and the diagnostic character of this variety is “Foliage blood-red” [27,28], the designation of a neotype was not simple since colours of amaranths usually change during the drying process. So, a coloured illustration (e.g., no. 227 published by Step & Bois [30]) would be desirable. Fortunately, I found just one specimen at GH (GH01928945) bearing the terminal part of a plant of *A. caudatus* with two leaves, of which one is clearly red-coloured. This plant was collected in America. GH01928945 is here designated as the neotype of *Amaranthus atropurpureus*.(2)*Amaranthus caudatus* var. *albiflorus*: the diagnostic characteristic given by Bailey [27] (p. 270), i.e., the colour of the flowers (“Spikes white or greenish white”), is very difficult to verify in specimens. In fact, as we know, colours of amaranths change after the exsiccation. I here designate a specimen preserved at RO (Figure 4) which was identified as “*Amaranthus caudatus* var. *albiflorus*” by Alfredo Cacciato, who was an expert of the genus *Amaranthus* in Italy in the 1970s. I studied most of Cacciato’s exsiccata during the last 15 years and I am sure that he referred to plants having white flowers (see e.g., [2,10]).(3)*Amaranthus caudatus* var. *gibbosus*: I tried to find a specimen collected in America (the native area of *A. caudatus*) whose morphology matches Bailey’s concept. Unfortunately, no specimen was found at either the main American herbaria (e.g., NY, PH, and US) or in some important European ones (e.g., BM, K, and P). Therefore, I was forced to choose from my own recent collection in Serbia (Eastern Europe) (Figure 5).

According to the treatment proposed in the present paper, *Amaranthus atropurpureus* and *A. caudatus* var. *albiflorus* are synonyms of *A. caudatus s.s.* (terminal florescence pendulous), whereas *A. caudatus* var. *gibbosus* is the new proposed name of *A. baileyanus* Iamonico, *nom. Nov.* (nodding synflorescence; see Section 3.3 Conclusions).

**Figure 4 plants-12-01566-f004:**
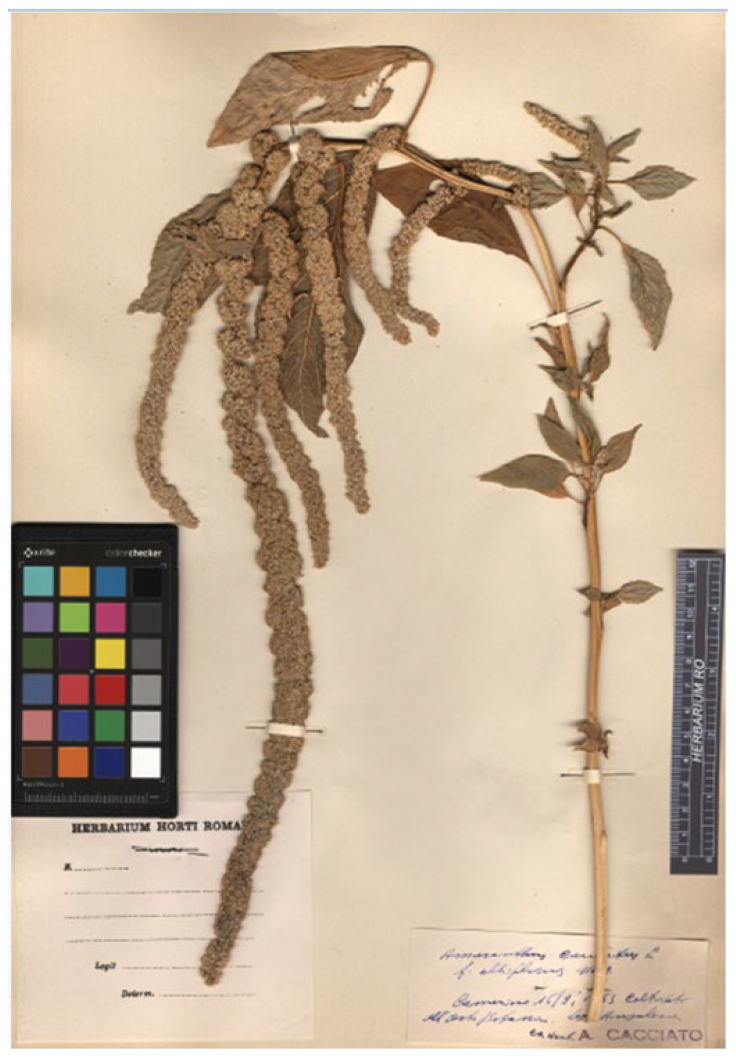
Neotype of the name *Amaranthus caudatus* var. *albiflorus* (RO!).

**Figure 5 plants-12-01566-f005:**
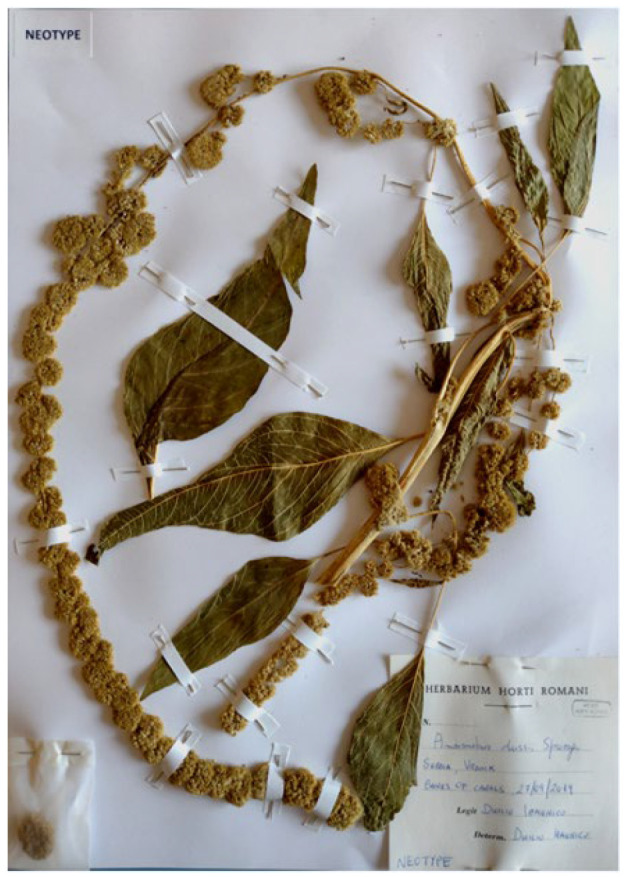
Neotype of the name *Amaranthus caudatus* var. *gibbosus* (RO!).

A further two notes, about two of the three Bailey’s varieties, are as follows:(1)Var. *gibbosus*: Bailey [31] (p. 252) published again (after Bailey [28]) the varietal name *gibbosus* by giving a very similar diagnosis (“interrupted spikes as if made up of separate heads or glomerules”). However Bailey, in his *Manual*, ascribed the varietal name to “Vilm.” (=“Vilmorin”), which was reported after the name and clearly refers to the surname of a famous French family of horticulturists [32]. According to the Bailey’s list “AUTORITIES FOR THE BINOMIALS” given in the first part of his *Manual* [31] (p. 41), it cannot be possible to understand to which member of this family the abbreviation “Vilm.” Refers. In fact, Bailey [31] (p. 41) reported “Vilm. Several generation of the family Vilmorin, Paris… Pierre Philippe André Leveque de Vilmorin, 1746–1804. Pierre Vilmorin, 1816–1860. Henry L. de Vilmorin, died 1899”. Note, however, that no Vilmorin’s reference was reported in Bailey’s *Manual* [31] (p. 252). As a consequence, the correct citation of this variety in Bailey’s *Manual* would be “var. *gibbosus* Vilm. ex Bailey” according to Art. 46.5 of ICN. Despite the difference in author citation (*Amaranthus caudatus* var. *gibbosus* L.H.Bailey in 1919 and *A. caudatus* var. *gibbosus* Vilm. ex L.H.Bailey in 1924), it is most probable that Bailey [31] (p. 252) just added “Vilm.” but referred to his previous published taxon [28] (p. 270). With the aim to verify if Bailey’s variety was previously published by one of the Vilmorins, I checked all the main online databases of plant names [33,34,35], but no *Amaranthus* name ascribed to one of Vilmorin’s was listed. I also checked all the works or papers in which Vilmorin published plant names (note that, according to the databases of plant names, nine persons of Vilmorin’s family were reported in the databases, i.e., P. V. L. de Vilmorin (1746–1840, abbreviated “V.Vilm.”), P. P. A. L. de Vilmorin (1776–1862, abbreviated “S.Vilm.”), P. L. F. L. de Vilmorin (1816–1860, abbreviated “Vilm.”), E. de Vilmorin (1826–1868, abbreviated “E.Vilm.”), C. P. H. L. de Vilmorin (1843–1899, abbreviated “H.Vilm.”), A. L. M. L. de Vilmorin (1849–1918, abbreviated “M.Vilm.”), J. M. P. Lévêque de Vilmorin (1872–1917, abbreviated “P.Vilm.”), J. L. de Vilmorin (1882–1933, abbreviated “J.Vilm.”), R. M. V. P. L. de Vilmorin (1905–1980, abbreviated “R.Vilm.”) with 0, 1, 60, 142, 1, 4, 1, 0, and 2 names, respectively. According to the mentioned databases, Vilmorin’s works which were published before 1924 (the year of the propologue of Bailey’s variety [31] (p. 252)) are as follows: volume no. 1 of *Revue horticole* (1843), volume no. 8 of *Annales des Sciences Naturelles* (1857), *Le Bon Jardinier* (year 1860), *Les plantes potagères* (1883), volume no. 83 of *Rad Jugoslavenska Akademije Znanosti i Umjetnost* (1887), volume no. 4 of *Garden and forest; a journal of horticulture, landscape art and forestry* (1891), *Vilmorin’s Blumengärtnerei* (1894), volume no. 16 of *Journal of the Royal Horticultural Society* (1894), volume no. 35 of *Journal of the Arnold Arboretum*, *Fruticetum Vilmorinianum* (1904), volume no. 52(6) of *Bulletin de la Société Botanique de France* (1906), *Mitteilungen der Deutschen Dendrologischen Gesellschaft* (1909), *Catalogue de graines de plantes de serre et d’orangerie* (1912–1913), and the journal of the *Société nationale d’horticulture de France* (1914). After checking all these works, I could verify that no var. *gibbosus* was published neither by P. F. A. Levêque de Vilmorin nor by P. L. F. L. de Vilmorin. The citation “*Cat Grain. Conif. Mexiq.*”, as reported in IPNI [33] for *Pinus otteana* Roezl ex Vilm., refers to Roezl’s *Catalogue des graines de Coniferes mexicains* (1857), and so to gymnosperms. As a consequence, I can state that the var. *gibbosus* was published in Bailey’s *Manual* [28] (p. 252) for the first time.(2)Var. *albiflorus*: Bailey’s trinomial is a later and illegitimate homonym of a Moquin-Tandon’s name which was published in Candolle’s *Prodromus* [24] (p. 256) (Art. 53.1 of ICN).

#### 3.1.5. *Amaranthus dussii*

*Amaranthus dussi* was honoured by Sprenger [36] (p. 178) to French Father Dussi who lived in Martinique and often sent plants to C. Sprenger; a description was given on the basis of plants growing in the Botanical Garden of Naples (Southern Italy) from seeds collected in Martinique (Lesser Antilles).

Carl Ludwig Sprenger was a German botanist (30 November 1846–13 December 1917) who lived in Naples from 1877 to 1907 where he was partner in the horticultural house of Damman & Co. of San Giovanni Testuccio (a district of the eastern area of Naples city). Sprenger collected many seeds and prepared hundred specimens which, however, were destroyed after the eruption of Vesuvius on 4 April 1906 [37] (p. 268). Original material for *Amaranthus dussi* is not, therefore, in extant and, according to Art. 9.8 of ICN, a neotypification is required. On the basis of the original description [36] (p. 178), A. dussi displays synflorescence with “fiori riuniti in lunghe e grosse spighe conglomerate prime erette e poi elegantemente riflesse e pendule” (=“flowers arranged in long and big spikes ammassed before erect, then stylishly reflexed and pendulous”). This trait is typical of just one Amaranthus species, i.e., *A. caudatus* [1,2,23]. I here propose a specimen preserved at NAP (barcode NAP0000610), collected in Naples Province, as the neotype of the name *Amaranthus dussi* (Figure 6). Based on Sprenger‘s description [36] (p. 178), and the treatment here proposed, *A. dussi* is to be considered as a synonym of *A. caudatus*.

#### 3.1.6. *Amaranthus edulis sensu stricto*

*Amaranthus edulis* was validly published twice, by Moquin-Tandon [24] (p. 277, as “*Amaranthus edulis* Michx.”) and Spegazzini [38] (p. 163). Moquin-Tandon’s name is not valid, since it was listed as a synonym of the legitimate *Acnida cannabina* L. var. *lanceolata* Moq. (Art. 36.1a of ICN). Therefore, Spegazzini’s name, despite being published later than Moquin-Tandon’s one (1917 vs. 1849), is legitimate and not a later homonym. Moquin-Tandon’s *A. edulis*, referring to *Acnida cannabina* var. *lanceolata* (and not to *A. caudatus*), is not reported in the taxonomic treatment of the present paper.

Spegazzini [38] (p. 163) provided a detailed diagnosis and description, as well as the provenance (“*Hab.* Cultivado en la región árida y montañosa de la provincia de Salta por la población indigena”). Bayón [6] (p. 276) indicated the holotype for this name (“TIPO: cultivado en La Plata, s.f., C. L. Spegazzini s.n. (holotipo, LPS-12665 en LP-16325!)”). However, first, no holotype was cited in the protologue and a lectotypification is necessary according to the Arts. 9.1, 9.3, and 9.4 of ICN (see also the considerations by McNeill [39]). So, Bayón’s holotype’s indication has to be considered as a lectotype. However, according to Art. 7.11 of ICN, “designation of a type is achieved only if… on or after 1 January 2001, if the typification statement includes the phrase “designated here” (hic designatus) or an equivalent”. Since this phrase was not reported by Bayón [6] (p. 276), his typification is not valid. I here designate the specimen LP12665 (cited by Bayón [6] (p. 276)) as the lectotype of the name *Amaranthus edulis* (Figure 7).

The lectotype at LP is identifiable as *Amaranthus caudatus s.lat.* on the basis of the shape of the tepals which are spatulate with obtuse apexes and, according to the treatment proposed in the present paper, as *A. mantegazzianus* Passer., having erect terminal florescence.

#### 3.1.7. *Amaranthus edulis* var. *spadiceus*

The var. *spadiceus* was proposed by Hunziker [40] (p. 330) to describe forms of *Amaranthus edulis* with light brown seeds and robust and longer bracts (“*Episperma spadiceo. Bracta larga et robusta, usque ad 3.6 mm, cum nervo et arista incrassatis*”); one specimen is listed (“Tucumán: Colalao del Valle, depart. Tafí leg. Hunziker, 23-III-1943 (A. T. H. n° 2552. *Typus varietatis*”)), where “A. T. H.” (Armando Theodoro Hunziker) refers to his personal herbarium as indicated in the section “Material and methods” by the author (“Además del material que guardo en mi colectión (A. T. H.)…” = In addition to the material that I keep in my collection (A.T. H.)…”). The above cited collection was found at CORD, where Hunziker’s collection is preserved, and it is the holotype (Figure 8). This CORD specimen is identifiable as *A. caudatus s.lat.* on the basis of the shape of the tepals, spatulate with obtuse apexes, and according to the treatment proposed in the present paper, as *A. mantegazzianus* Pass., having erect terminal florescence. The diagnostic characteristics given in the protologue (seed colour, length, and structure of the bracts) has no taxonomic value (see, e.g., [1,2,6,22,23]), and this variety name is synonymized with *A. mantegazzianus*.

#### 3.1.8. *Amaranthus mantegazzianus*

*Amaranthus mantegazzianus* was proposed by Passerini [41] (p. 4) on the basis of plants cultivated at the Botanical Garden of Parma (Parma is a city of the Emilia-Romagna region, Northern Italy) from seed collected in Argentina (Province of Salta). The diagnosis is as follows: “caule erecto angulato glabro, apice pubescente, viridi, deio, praesertim superne, luteo-fulvo; foliis petiolatis, ovato-oblongis, acuminatis viridibus glabris, paniculis amplis subcorymbosis, spicis crassis obtusis, lateralibus demum cernuis; floribus densis badio-fulvis, calyces bracte subaequante, sepalis membranaceis obovatis apice aristulatis; utriculis badiis ovato-trigoni; apice bi-tricuspidatis, seminibus albus orbicularibus margine tumidiusculis”.

Hunziker [40] (p. 330) designated a neotype for Passerini’s name on a specimen collected in Salta Province (CORD00002607; Figure 9); isoneotypes (at K, SI, and US) were also reported. Since Hunziker [40] (p. 330) did not cite the herbarium Parma, where Passerini’s collection is preserved, I tried to check this herbarium, but unfortunately no original material was traced (R. Brusi pers. Comm.). As a consequence, Hunziker’s choice is correct, and it is to be accepted.

#### 3.1.9. Illegitimate and Invalid Names

The names *Amaranthus pendulinus* and *A. pendulus* were reported by Moquin-Tandon [24] (p. 256) in Condolle’s *Prodromus* as synonyms of *A. caudatus* var. *albiflorus*. These two names were not validly published according to Art. 36.1a of ICN.

Bailey [31] (p. 252) listed the name “*Amaranthus abyssinica*” as synonym of *A. caudatus*. According to Art. 36.1a of ICN, Bailey’s name is not validly published.

Iamonico [21] (p. 110 in Table 5), in his work on Moquin-Tandon’s *Amaranthus* names, inadvertently published the name “*Amaranthus caudatus* var. *parviflorus* Moq.”. However, this variety was never published by Moquin-Tandon (1849: 256) under *A. caudatus*, who validly described an *A. albus* L. var. *parviflorus* Moq. The name, as reported by Iamonico [21] (p. 110 in Table 5), is to be considered as a nomen nudum, and, therefore, invalid according to Arts. 38.1 and 38.2 of ICN.

### 3.2. Taxonomic Notes

*Amaranthus caudatus* was validly published in the first edition of *Species Plantarum* [42] (p. 990) and correctly typified on a Linnaean specimen (Herb. Linn. 1117.26) by Townsend [43] (p. 10). This species is currently accepted by the scientific community, and it morphologically differs from the other monoecious *Amaranthus* taxa by the following sexual characteristics: terminal, lax, pendulous (especially the terminal one), erect, or nodding, and very long (up to 80 cm) often red or purple synflorescences; five spatulate-obovate tepals, equal or subequal to the bracts; and dehiscent fruit.

On the basis of the ongoing studies on the genus *Amaranthus*, I was able to note that *Amaranthus caudatus*, although less than other monoecious amaranths (e.g., *A. retroflexus* L. or *A. hybridus* L. (see, e.g., [1,2,6,23]), displays a phenotypic variability, especially in the synflorescence structure which can be erect, pendulous (especially the terminal florescence), or nodding (Figure 10). These morphotypes are referable to *A. mantegazzianus*, *A. caudatus s.s.*, and *A. caudatus* var. *gibbosus*, respectively. Moreover, there is also cytological and molecular evidence which allows to distinguish these three taxa. *A. caudatus s.s.* and *A. mantegazzianus* have 2n = 32 [22,44,45,46,47,48,49,50,51,52,53,54,55,56,57], whereas the taxon gibbosus shows 2n = 30 [49]. *A. caudatus* and *A. mategazzianus* are, in turn, different by the chromosome asymmetry index (0.2491 and 0.3701, respectively) and the DNA content (2C = 1.35 ± 0.013 and 1.46 ± 0.015, respectively; see [52,53,54,58]) and the distribution and variability of constitutive heterochromatin [56].

### 3.3. Conclusions

On the basis of morphological, cytological, and molecular data, the taxa caudatus, mantegazzianus, and gibbosus deserve to be treated as separate species, as proposed below. A new combination would be necessary for Bailey’s var. gibbosus. However, note that an *Amaranthus gibbosus* was already and validly published by Bailey [27] (pp. 55–56) (diagnosis: “pigweed and beet-roots”), and a new combination of the var. *gibbosus* by Bailey [28] (p. 270) would result as a later homonym and illegitimate name (Art. 53.1 of ICN). As a consequence, a new name is proposed here.

### 3.4. Taxonomic Treatment

Images of the types which are available online and not published in the present manuscript are reported in Appendix A.

***Amaranthus caudatus*** L., Sp. Pl. 1: 990. 1953 ≡ *Amarnathus hybridus* L. subsp. *Caudatus* (L.) Iamonico & Galasso, Italian Botanist 4: 34. 2017.

Lectotype (designated by Townsend [43] (p. 10)): Herb. Linn. 1117.26 (LINN (image!); Appendix A).

= *Amaranthus maximus* Mill., Gard. Dict., ed. 8: Amaranthus 5. 1768 ≡ *Amaranthus caudatus* var. *maximus* (Mill.) Moq., Prodr. (DC.) 13(2): 256. 1849.

Lectotype (designated by Iamonico [11] (p. 65, Figure 1)): United Kingdom, London, Chelsea Physic Garden, 1741, *s.c. 954* (BM000832631 (image!); Appendix A).

= *Amaranthus caudatus* L. var. *albiflorus* Moq., Prodr. (DC) 13(2): 255. 1849.

Lectotype (designated by Iamonico [12] (p. 93)): Switzerland, Hort. Genev., 1840, *A.P. Candolle 397* (P04021950!; Appendix A).

= *Amaranthus caudatus* L. var. *alopecurus* Moq., Prodr. (DC) 13(2): 256. 1849 ≡ *Amaranthus alopecurus* (Moq.) Hochst. ex A.Br. & al. (not “*Amaranthus alopecurus* Hochst. ex. A.Br. & D.C.Bouché” as reported by the online databases of plant names).

Lectotype (designated by Bajón [6] (p. 276)): Ethiopia, In ruderatis prope Adoam, 1 November 1844, *A.F.W. Schimper 1535* (P00482809 (image!); Appendix A). Isolectotypes: BR0000008357557 (image!) (Appendix A), GH00037040 (image!); Appendix A), HAL0110480 (image!) (Appendix A), HOH009263! (Appendix A), K000223569 (image!; the collection number was erroneusly reported (as “1537”) in the online K catalogue) (Appendix A), K000223570 (image!; the collection number was erroneusly reported (as “1537”) in the online K catalogue), exsiccata on the left (Appendix A), MEL2459427 (image!) (Appendix A), MO357985 (image!) (Appendix A).

= *Amaranthus dussii* Spreng., Bull. Soc. Tosc. Ortic. 21: 178. 1896.

Neotype (designated here): Italy, Campania region, Naples Province, Ischia island, 5 October 1847, *s.c. s.n.* (NAP0000610!; Figure 6).

= *Amaranthus caudatus* var. *albiflorus* Vilm. Ex L.H.Bailey, Stand. Cycl. Hort.: 270. 1919, *nom. Illeg.* Non Moq. (Art. 53.1 of ICN).

Neotype (designated here): Italy, Marche region, Camerino town, *all’Orto Botanico*, 16 September 1965, *B. Anzalone* (*ex herb.* A. CACCIATO) (RO!; see Figure no. 20 in Iamonico 2015a: 46; Figure 4 in the present paper).

= *Amaranthus caudatus* var. *atropurpureus* L.H.Bailey (citation according to Art. 58.1-Ex.3 of ICN), Stand. Cycl. Hort.: 270. 1919 ≡ *Amaranthus atropurpureus* L.H.Bailey, Cycl. Hort.: 55. 1909, *nom. Illeg.* (Art. 53.1 of ICN) *non A. atropurpureus* Roxb., Fl. Ind. III: 608. 1832.

Neotype (designated here): U.S.A., Virginia, Roland; 2 miles N.W. of Thoroughfare Gap., S.W. base of Bull Run Mts., planted in small garden in weedy field, 09 October 1978, *N. A. Harriman* (GH01928945 (image!); Appendix A).

= *Amaranthus caudatus* L. subsp. *Saueri* V.Jehlík, Preslia 62: 164. 1990.

Holotype: Germany, Bohemia, in horto facturae in vico Podhuri prope opp. Vrehalbí culta (= im Fabrksgarten in Harta), 435 m s.m., 25 October 1923, V. Cypers s.n. (PR615740 (image!)). (Figure 1).

– *Amaranthus pendulinus* Moq., Prodr. (DC) 13(2): 255. 1849, nom. inval. pro synonym of *A. caudatus* var. *albiflorus* (Art. 36.1a of ICN).

– *Amaranthus pendulus* Moq., Prodr. (DC) 13(2): 255. 1849, nom. inval. pro synonym of *A. caudatus* var. *albiflorus* (Art. 36.1a of ICN).

– *Amaranthus abyssinicus* L.H.Bailey (as “*abyssinica*”), Man. Cult. Pl.: 252. 1924, nom. inval. pro synonym of *A. caudatus* (Art. 36.1a of ICN).

*Native distribution area*. The origin of *Amaranthus caudatus* remains uncertain at the current state of knowledge. According to several authors (e.g., [1,2,23,59]), this species most likely originated in South America (Argentina, Equador, Perù, and Bolivia) by domestication and crossing with the wild *A. quitensis* Kunth.

*Current distribution area.* According to the current available data, *Amaranthus caudatus* would occur currently as alien species in Asia [60], Australia [59,61], Europe [62], and Africa [63]. However, it cannot be possible, at present, to confirm the occurrence of this species at the national level for the following reasons:(1)The name *A. mantegazzinus* was rarely cited and accepted as separate taxon over time. Sometimes, it was indicated in a note under *A. caudatus* (see e.g., [23]), whereas in other cases it was synonymized with the Linnaean name (see e.g., [34]).(2)In some cases (e.g., [63]), *A. quitensis* is reported as heterotypic synonym of *A. caudatus*.(3)*Amaranthus caudatus* var. *gibbosus* (≡ *A. baileyanus* Iamonico, *nom. nov.*, see below) was rarely indicated after Bailey [28].

Further investigations (filed surveys and herbarium examinations) will be necessary to provide a distribution of *Amaranthus caudatus* out of its native range.

*Selected specimen examined*. Bolivia: Hacienda Simaco sobre el camino a Tipuani, 1920, *Buchtien 5402* (US03541823). Bosnia-Herzegovina: Zivinice, 215 m a.s.l., 30 September 2020, *S. Sarie, s.n.* (RO). Chile: Santiago, 1918, *Claude-Joseph 712* (US03541811). China: Xizang; Tíbet. Province: Bálti. Environs of Skárd, *s.d.*, *Schlagintweit s.n.* (US03542416). Italy: Emilia-Romagna, inselvatichico nelle vicinanze di Bologna, July 1886, *Mattei s.n.* (FI); Liguria: Varazze, orticolo?, 10 October 1929, *Gresino s.n.* (FI!); Piemonte, Trontano, Quarata, campo, 248 m a.s.l., 18 September 2002, *Antonietti s.n.* (*Herb. Antonietti*!, RO). Lybia: Cyrenaica, El Hamrah, 15 December 1873, *Ascherson 2064* (M0241385). India: Chickpet, Karnataka, 320 m a.s.l., 21 December 2021, *Arya Sindhu,* 675 (UCBD25). Netherlands: *s.d.*, Clifford *s.n.* (BM000647396). Peru: Lambayeque; Dep. Lambayeque, Prov. Chiclayo, Camino a San José, April 1951, *López 0290* (US03541813). Romania: Oravita, 215 m a.s.l., 9 January 2019, *Iamonico s.n.* (RO). Serbia: Kragujevac, artificial habitat, 356 m a.s.l., 9 February 2019, *sin coll., s.n.* (RO). Switzerland: *Hort. Genev.*, 1840, *Candolle 397* (P04021950). U.S.A.: Illinois, Chicago. 3311 North Seeley Ave, 13 July 1987, *T. C.Plowman 14507* (US03540303).

***Amaranthus baileyanus*** Iamonico, *nom. nov. pro Amaranthus caudatus* var. *gibbosus* L.H.Bailey, Stand. Cycl. Hort.: 270. 1919, *non A. gibbosus* L.H.Bailey, Cycl. Hort.: 55–56. 1909.

Neotype (designated here). Serbia, Vrdnik, banks of canals, 27 September 2019, D. Iamonico s.n. (RO!; Figure 5; isoneotype NY (image!)).

*Etimology*. The specific epithet is dedicated to L. H. Bailey, who was the author of the basionym.

*Native distribution area***.** Unknown, but likely North America. *Amaranthus baileyanus* was, in fact, originally described from plants cultivated in North America (see Bailey 1919: v) that “grown within its territory [North America] which are now subject of living interest or likely to be introduced…” [28] (p. vi).

*Current distribution area***.** No data about *Amaranthus baileyanus* appear to be published. On the basis of my proposed neotypification, this species occurs in Eastern Europe (Serbia), where I directly saw a population in the field (collection was here designated as the neotype of *A. baileyanus*). Moreover, I traced two specimens from France collected more than 80 years ago (see the following “Selected specimens examined”). The species is here considered as a casual alien for Europe. Further investigations will be necessary to provide data on the chorology of *A. baileyanus*.

*Selected specimens examined*. France: Puy-de-Dôme, September 1936, *Ch. D’Alleizette s.n.* (CLF153172, image available at http://mediaphoto.mnhn.fr/media/1444837906403HZ79dhxGDbFYMQ1Y; accessed 3 April 2023); Val-d’Oise, Maffliers, 27 October 1940, *M. P. Jovet s.n.* (P02602557, image available at http://mediaphoto.mnhn.fr/media/1526301113151B95P7QdjOye5watp; accessed 3 April 2023).

***Amaranthus mantegazzianus*** Passer., Hor. Parm.: 4. 1865 ≡ *Amaranthus caudatus* L. subsp. *mantegazzianus* (Passer.) Hanelt, Kulturpflanze 16: 128 1968.

Neotype (designated by Hunziker [64] (p. 105)): Argentina, Tacumán, Dep. Tafí, Colalao del Valle, 23 March 1943, *A.T. Hunziker 2555* (CORD00002607 (image!), Figure 9; Appendix A). Isoneotypes (indicated by Hunziker [64] (p. 105)): K000582941 (image!) (Appendix A), SI00718 (image!) (Appendix A), US00106250 (image!) (Appendix A).

= *Amaranthus edulis* Speg., Physis (Buoenos Aires) 3: 163. 1917.

Lectotype (designated here): Argentina, cultivado en la Plata, *s.d.*, *C. Spegazzini s.n.* (LP002715 (image!), Figure 7; Appendix A).

= *Amaranthus edulis* Speg. Var. *spadiceus* Hunz., Revista Argent. Agron. 10: 330. 1943.

Holotype. Argentina, Tafi, Tucuman, “Cultivado por su semillias alimenticias. De + 1.50 m de altura”, s.d., A.T. Hunziker 2552 (CORD00009356 (image!); Figure 8).

= *Amaranthus edulis* Speg. Var. *pseudopaniculatus* f. *pseudopaniculatus* Suessenguth in Suessenguth & Merxmüller 1951: 71, Mitt. Bot. Staats., Munchen 1: 71. 1951.

Holotype. Tanzania, Amani, 2900 ft., 24 March 1941, P.J. Greenway 993 (EA no. 6154 (image!); Figure 2).

= *Amaranthus edulis* Speg. Var. *pseudopaniculatus* f. *oblongipetalus* Suessenguth in Suessenguth & Merxmüller 1951: 71, Mitt. Bot. Staats., Munchen 1: 71. 1951 (as “*oblongopetalus*”; see Art. 60.10 of ICN).

Holotype. Tanzania, Amani, 2900 ft., 24 March 1941, P.J. Greenway 995 (EA no. 6155 (image!); Figure 3); isotype K000195694! (Appendix A).

*Native distribution area*. Unknown, but likely South America (Argentina).

*Current distribution area***.** The holotypes of *Amaranthus edulis* var. *pseudopaniculatus* (both f. *pseudopaniculatus* and f. *oblongipetalus*) came from Tanzania, whereas the specimens below listed were from Ethiopia (they are the types of *A. caudatus* var. *alopecurus* Moq., which was considered by Iamonico [12] as a synonym of *A. caudatus s.s.*)). I here consider *A. mantegazzianum* as a probably alien species (casual) for Africa. Further investigations will be necessary to give a distribution of *Amaranthus mantegazzianus* out of its native range.

*Selected specimen examined*. Ethiopia: *In ruderatis prope Adoam*, 1 November 1844, *Schimper 1535* (P00482809); *ibidem* (BR0000008357557, GH00037040, HAL0110480, HOH009263, K000243571, MO357985).

## Figures and Tables

**Figure 1 plants-12-01566-f001:**
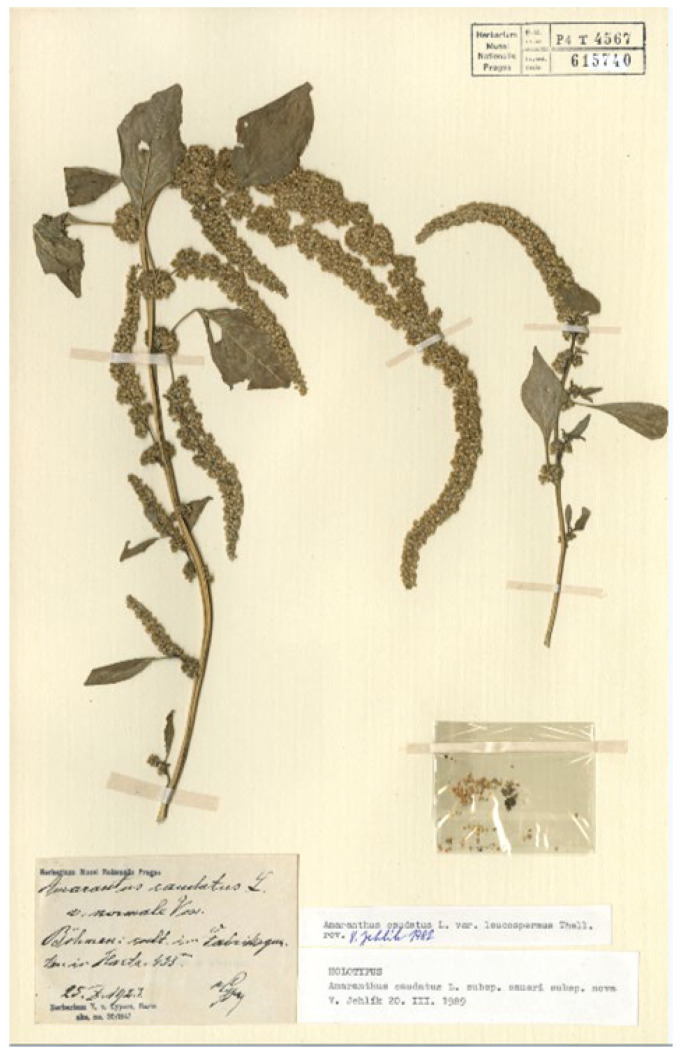
Holotype of the name *Amaranthus caudatus* subsp. *saueri* (PR615740!).

**Figure 2 plants-12-01566-f002:**
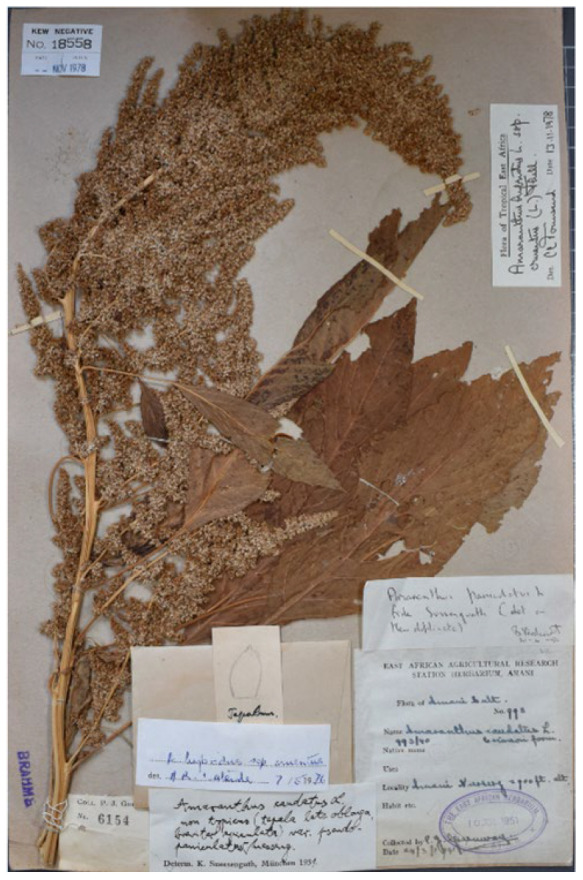
Holotype of *Amaranthus caudatus* var. *pseudopaniculatus* f. *pseudopaniculatus* (EA no. 6154).

**Figure 3 plants-12-01566-f003:**
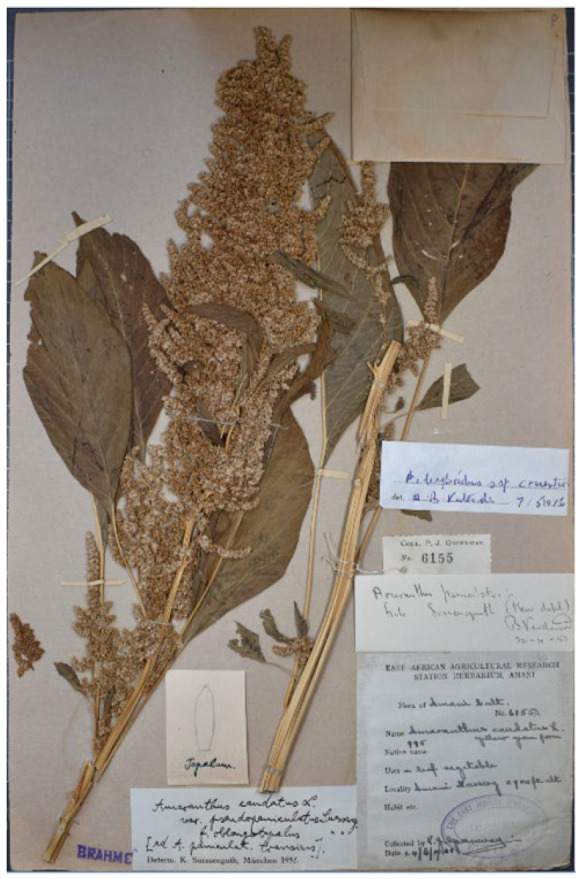
Holotype of *Amaranthus caudatus* var. *pseudopaniculatus* f. *oblongipetalus* (EA no. 6155).

**Figure 6 plants-12-01566-f006:**
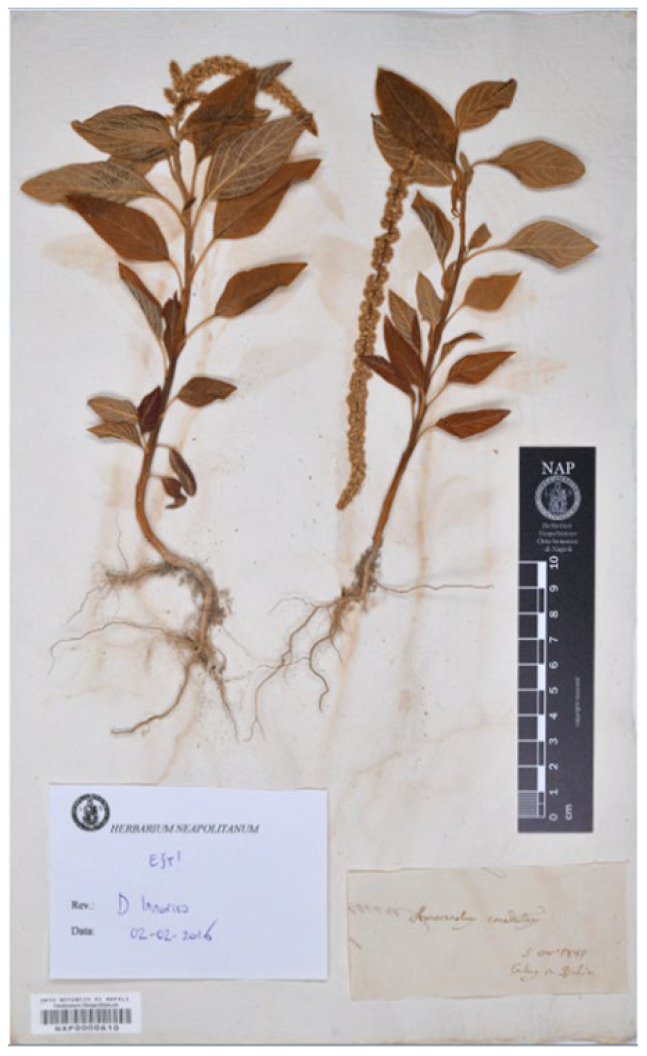
Neotype of the name *Amaranthus dussii* (NAP0000610!).

**Figure 7 plants-12-01566-f007:**
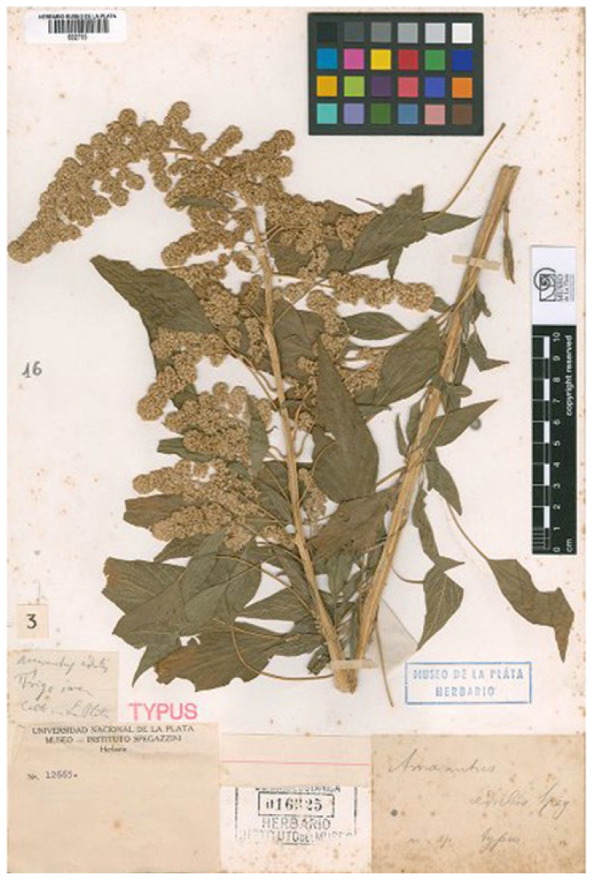
Lectotype of the name *Amaranthus edulis* var. *edulis* (LP002715 (image!)).

**Figure 8 plants-12-01566-f008:**
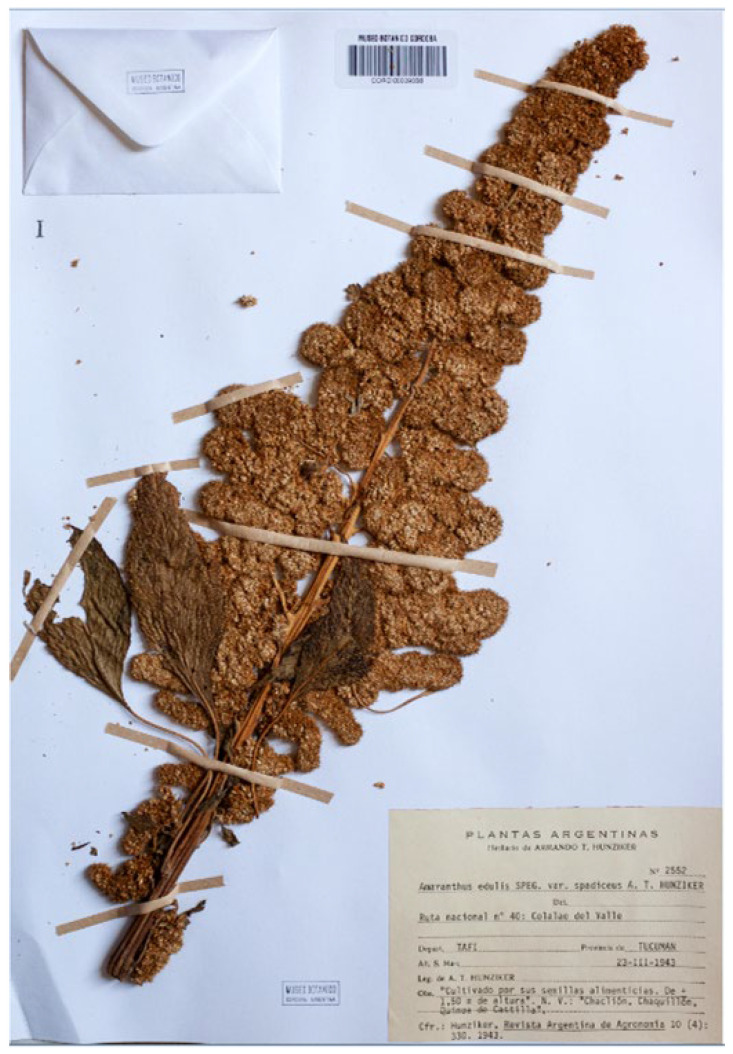
Holotype of the name *Amaranthus edulis* var. *spadiceus* (CORD00009356).

**Figure 9 plants-12-01566-f009:**
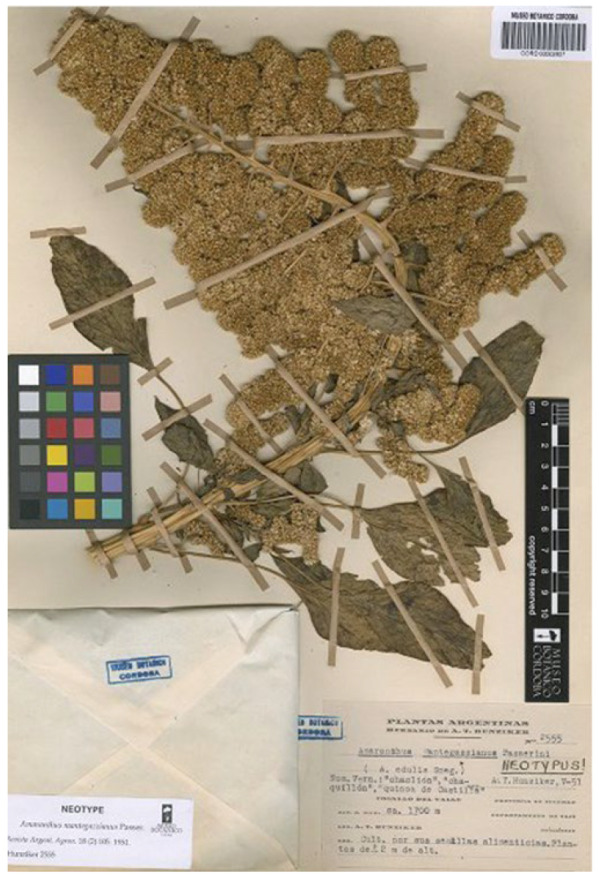
Neotype of the name *Amaranthus mantegazzianus* (CORD00002607).

**Figure 10 plants-12-01566-f010:**
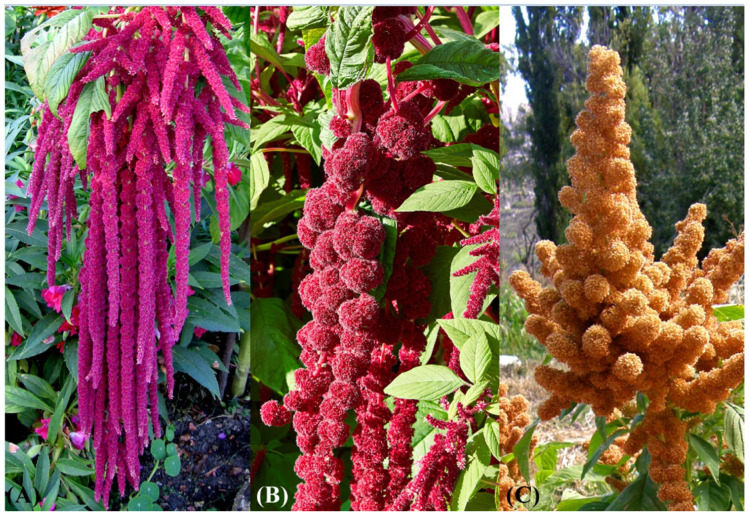
Structure of the synflorescences in the *Amaranthus caudatus* (**A**), *A. baileyanus* (**B**), and *A. mantegazzianus* (**C**). Photos modified from original images by the following authors: D. Biville (photographed at the Bergius Botanic Garden (Stockholm, Sweden) in 22 September 2006), all rights released, public domain (https://commons.wikimedia.org/wiki/File:Image_005_Amarante_Queue_de_renard.jpg?uselang=it; accessed 3 April 2023); C. T. Johansson (photographed at the Bergius Botanic Garden (Stockholm, Sweden) in 6 September 2015), Creative Commons Attribution 3.0 Unported license (https://commons.wikimedia.org/wiki/File:Amaranthus_caudatus-IMG_9189.jpg; accessed 3 April 2023); and Bachelot Pierre J.-P. (photographed at San Francisco de Tilcara (Argentina) in 25 March 2012), Creative Commons Attribution-Share Alike 3.0 Unported, 2.5 Generic, 2.0 Generic and 1.0 Generic license (https://commons.wikimedia.org/wiki/File:Amaranthus_mantegazzianus.JPG; accessed 3 April 2023).

## Data Availability

Not applicable.

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
