# Peer review of "Nomenclature Survey of the Genus *Amaranthus* (Amaranthaceae): 12 Questions about *Amaranthus caudatus"

_plants, 2023, doi:10.3390/plants12071566_

Round 1

Reviewer 1 Report

This paper endeavors to make progress on the nomenclature of Amaranthus by typifying several names. Substantial work has gone into this, and the goal is worthy. However, the paper is awkwardly written, with some questionable decisions or outright mistakes, so that major revision is needed before it can be published. The substantive questions regarding this paper are whether the typifications are needed, whether the types are well chosen, and whether the information is presented in a clear format. I do not care for the format, with lengthy discussion of names that have not yet been introduced with a nomenclature block. You might consider adopting a more traditional format, unless the journal requires this format.

Also, the paper includes some taxonomic decision-making. The methods section says that field and herbarium studies were carried out, and the “author contribution” says that the author perform[ed] statistic[al] analysis. But we see very little of the results. There’s no discussion of any numerical data. Specimens are not cited, distributions are not given, and there’s no clear discussion of the characters supporting each recognized species. The brief Taxonomic Notes, sect. 3.2, provides very helpful information some of which might better be included in the introduction, or in discussions of individual taxa. The morphology of “nodding” is not well distinguished by the photos (Fig. 11) and I have to wonder if any other characters support that species as distinct. Is more detailed taxonomic work going to be presented separately in another paper?

Previous typifications have been located and documented, so while I do not have access to many of the protologues, I presume the names identified as needing typification really do need it.

Under A. caudatus, there are two synonyms given neotypes from Italy, A. dussii (NAP, 1847) and A. caudatus var. albiflorus Vilm. ex Bailey (RO, 1965), and one neotypified on an illustration, A. caudatus var. atropurpureus Bailey (an 1897 illustration). Amaranthus caudatus is said to come probably from South America and to be widely distributed elsewhere now, though its presence cannot be confirmed at national level because of confusion with the two other species that the author wishes to segregate. None of the types chosen has any apparent connection with the work of the authors of the names. Why pick an 1847 specimen that has no duplicates anywhere, much less on the species’ continent of origin, when you could pick a modern collection that has five or ten duplicates?

Illustrations are not even allowed as types of new plant taxa anymore. Most people would argue against choosing an illustration as neotype unless there was a very good reason, either a connection to the name or a total lack of good herbarium material. You can’t, for example, ever get DNA out of an illustration. For var. atropurpureus, you don’t seem to have a good reason to pick this illustration, or you don’t explain it well. Why was the illustration not the type of var. albiflorus Vilm. ex Bailey, and the Italian specimen the type of var. atropurpureus? All seem to be arbitrarily chosen.

I have a suggestion for you: There is no requirement that you pick a different type for every name. If you are 100% sure that all these varieties should be forever lumped into the synonymy of A. caudatus, you could pick a good collection with duplicates and designate it as the neotype of ALL THREE, making them homotypic synonyms. If you are NOT sure, on the other hand, that these varieties will never be recognized as distinct, then it sounds like it would be premature for you to typify them, since you have not indicated that you recognize any morphological differences between them, so your types have been arbitrarily chosen and might not support future use of the names.

The neotype for A. caudatus var. gibbosus (≡ A. baileyanus, nom. nov., stat. nov.) is from Serbia, though the species is said to be probably from North America and to be only a “casual alien” in Europe. It is a recent collection by the author himself with one duplicate at RO. Minimally, I’d think it desirable to have collected three duplicates: one for Rome, one for a herbarium in the host country, and one to send to US or MEXU, or another large herbarium within what you consider to be the species’ native range. Is there really no such collection available?

The fifth typified name is A. edulis, another synonym of A. caudatus s.str. It is described as “designated by BAYÓN [6] (pag. 276) as ”holotype”, here corrected accord ing to the Art. 9.10” and the introductory paragraph about this name says the same. This is not an effective typification. While you should explain that Bayon called that specimen, which is presumed original material, a holotype, he did not effectively designate it as type, because his work was published in 2015, and from 2001 on, a designation must say “hic designatus”, “designated here”, or some similar phrase. Therefore you do not need to CORRECT his typification, which didn’t exist, but to designate the lectotype yourself - and since you also have not said “here designated”, yours also will not be effective!

Among the synonyms of A. caudatus are two publications of A. caudatus var. albiflorus. However, the later homonym, var. albiflorus Vilm. ex Bailey, has a parenthetical “as abyssinica”. In the invalid synonymy you list A. abyssinicus Bailey, not validly published four years later, which is irrelevant to the status of the varietal name. How can “abyssinica” be corrected to “albiflorus”? This requires some kind of explanation, unless you meant to type “albiflora” and “abyssinica” was just an error.

There is an orthographic issue that has been overlooked by previous authors. The epithet of A. caudatus fo. oblongopetalus, under Art. 60.10 of the ICN should be automatically corrected to “oblongipetalus.” In the main text, it should be spelled correctly, and in the nomenclature block, after the authority’s name a parenthetical “(as ‘oblongopetalus’)” should be added.

Another minor point is that the correct abbreviation for Bailey is “L.H. Bailey”, not just Bailey.

In the footnote at the bottom of p. 12, why is the invalid “A. edulis Michx.” referred to as “Moquin-Tandon’s name”? - And why is it not listed among “illegitimate and invalid names” on pp. 17-18?

In lines 107-110, the text is confusing, suggesting multiple specimens for each; it would be better to say “collection numbers and the herbarium in which each was deposited.” Later it should say “these quotations”. I do not have access to the protologue and wonder why there is a holotype for forma oblongopetalus if there are two cited specimens.

In lines 224-225 you say “any holotype was cited in the protologue” for a name that you are arguing did not have a holotype. It would be better to say “NO holotype was cited in the protologue (a locality alone not being sufficient)”.

The English is good, but not perfect. For example, in the beginning of the introduction, it would be better to say “half” than “the half”, and “medicines” rather than “medicals”; “pendolous” is used repeatedly for “pendulous.” In the second paragraph of the introduction, subgeneric names should be italicized. There are also rather numerous typos, e.g., in the last line of the abstract “vat.” for “var.”; in Sect. 3.1.4 “si”, “identify with” and “Bayley’s”; the genus name is misspelled in at least two different ways. I saw many more, but can’t provide corrections on a PDF and suggest that careful proofreading be done during revisions.

The format of the nomenclature blocks, with very lengthy web addresses for the images of numerous duplicates, is almost unreadable. I suggest omitting the URLs. Providing the herbarium and barcode will allow anyone who wants to see the images to do so easily by searching in the online databases (for those who have access to JSTOR Plants - and for those who do not, the URLs are a pointless burden). Some URLS have been needlessly lengthened by inclusion of query data; see that for MEL2459428, which could be truncated before the question mark without any loss of function, and that of the type of A. edulis. Also, adding not just “image at” but for each name and type status beginning with “image of the [iso]lectotype is available at” adds useless text to the blocks.

On top of all those URLs, ten type images are currently presented in the manuscript as full-page illustrations. Unless the journal wants this to be a giant manuscript because they’re short of manuscripts this month, I suggest reducing the type images to ¼ page so that they’d only take up 2.5 pages instead of 10. The different inflorescence shapes could be seen at that size, and other tiny characters supporting species identity can’t be seen at whole-page size anyway.

In the keywords, “A. mantegazzianus” isn’t a good keyword; if you want it to be searchable, spell it out. Finally, I think citation of all of the author’s previous nomenclatural publications on Amaranthus is a bit excessive.

Author Response

Dear reviewer,

thanks for the suggestions given. The paper was corrected according to the most of them. A Word file with Track Change on is attached. Further, another Word file with replies to comments is provided.

Best regards,

Duilio Iamonico

Reviewer 2 Report

Title: Nomenclature Survey of the Genus Amaranthus (Amaranthaceae). 12. Questions about Amaranthus caudatus

Author: Duilio Iamonico

This is an interesting, and well-written manuscript in which several taxa of the genus Amaranthus (Amaranthaceae) are correctly typified, and a new name (Amaranthus bayleyanus) is proposed.

            Abstract, Keywords, and Literature cited are pertinent and up-to-date. Typification section is, in general, adequate and well documented.

The author of the manuscript is a well-known expert in taxonomy and nomenclature of the genus Amaranthus.

Therefore, I recommend the publication of this manuscript and consider it appropriate to be published in the journal Plants though this last aspect should be, obviously, decided by the editor.

Some aspects of the manuscript should be also revised or modified:

a) Pages 7 and 21, Fig. 5. I have doubts about the choice of an alien plant collected in Serbia as neotype of Amaranthus baileyanus when this taxon is of American origin. In my opinion it would be better to choose a specimen from a native American population as the neotype.

b) Pages 7, 11, and 21. There is conflicting information in the manuscript about the protologue of Amaranthus caudatus var. gibbosus.

Some other minor points that should be corrected are marked in red in the manuscript.

Santiago Ortiz

Author Response

Thank you for your helpful comments, please see the attached file for our responses.

Round 2

Reviewer 1 Report

The manuscript is improved in a number of ways, but a couple of new problems have been introduced.

Under sect. 3.1.4, "varieties described by Bailey", you now speak of typifying not A. caudatus var. atropurpureus, but A. atropurpureus L.H. Bailey. In the nomenclature block, you refer to the variety as "var. atropurpureus (L.H. Bailey) L.H. Bailey", though A. atropurpureus L.H. Bailey is noted to be illegitimate as a later homonym. Since an illegitimate name cannot serve as a basionym, var. atropurpureus must simply be credited to L.H. Bailey. It could be viewed either as a new taxon (in which case their types could be different), or a "nomen novum" that simply happens to re-use the same epithet. The latter allows both to automatically share the same type and is the preferable view here.

The name of forma oblongopetalus was noted to be orthographically incorrect; it should be corrected to oblongipetalus. That is now correctly represented in the nomenclature block. However, text discussions throughout the paper need to use correct orthography. Much of sect. 3.1.3 uses "oblongopetalus", and the first paragraph of that section, and the abstract, both use "oblongitepalus" - tepal, not petal.

You have changed an illustration neotype to a red-leaved specimen in GH, which you say you had to look for. It's marked "!", indicating that it was seen. Did you actually see this specimen sometime earlier, or have you observed it only as an image? If the latter, the same question could be asked of other specimens so marked. I am not an expert on Amaranthus but to me species can be difficult to identify and the use of a dissecting microscope often seems necessary, so I wonder if you are really able to critically identify a rather fragmentary specimen just from a photo. You should not designate a type that you have not directly observed unless there is absolutely no possible doubt as to its identity.

Author Response

Thank you for your helpful comments, please see the reply attached.